

# Seismic soil-structure interaction analysis of wind turbine support structures using augmented complex mode superposition response spectrum method

Masaru Kitahara[1], Takeshi Ishihara[1]

[1]Department of Civil Engineering, School of Engineering, The University of Tokyo, 7-3-1, Hongo, Bunkyo-ku, Tokyo, Japan

*Correspondence to*: Takeshi Ishihara (ishihara@bridge.t.u-tokyo.ac.jp)

**Abstract.** In this study, the seismic soil-structure interaction (SSI) of wind turbine support structures is investigated using response spectrum method (RSM) based on the complex eigenmodes. Seismic loadings on wind turbine support structures

are newly derived by complex mode superposition RSM. To improve the prediction accuracy of the shear force acting on footings, this method is augmented by introducing the upper limit of modal damping ratios of 10 %. In addition, the bending moment at the hub height due to the mass moment of inertia of rotor and nacelle assembly is considered as an additional loading. The proposed method is validated by comparison with time history analysis (THA) accounting for different types of foundations and different tower geometries. Seismic loadings acting on the towers and footings by the proposed method

show favourable agreement with the mean results by THA of several input acceleration time histories, while the original complex mode superposition RSM strongly underestimates shear forces acting on footings.

## 1 Introduction

In recent years, the expansion in wind energy has increased the construction of wind turbines in seismically active regions, including Japan, and damages on wind turbine support structures caused by huge earthquakes have been reported. A piled

foundation was damaged at Kashima wind farm during Great East Japan Earthquake (March 11, 2011) (Ashford et al., 2011) and a wind turbine tower was buckled at Kugino wind farm during Kumamoto Earthquake (April 16, 2016) (Harukigaoka Wind Power Inc., 2016). In order to assure the structural integrity of wind turbine support structures against such huge earthquakes, an accurate and efficient method to estimate seismic loadings on wind turbine support structures is thus desired to be developed.

Response spectrum method (Der Kiureghian, 1981; Chopra, 2011) has been widely incorporated into the codes for the seismic analysis and design of various types of structures, including bridges, high-rise buildings, and nuclear power plants, due to its simplicity and efficiency (Eurocode 8, 2006; the Building Standard Law of Japan, 2004; American Society of Civil Engineers (ASCE), 2006). However, these codes are not directly applicable to wind turbine support structures owing their unique structural characteristics. Wind turbines have very low structural damping compared with the above other structures.



The response spectra of the structures with low damping ratios show large fluctuations, whereas damping correction factors in the above codes cannot accurately capture the uncertainty in the response spectra due to such fluctuations. To cope with this issue, Ishihara et al. (2011) have proposed a new damping correction factor for wind turbine support structures, in which the uncertainty in the response spectra is considered by employing a quantile value. Moreover, Kitahara and Ishihara (2020) have recently extended it to be applicable to MW class large sized wind turbine support structures as well.

Moreover, the mass ratio between the super- and sub-structures of wind turbines is very different, and the footing mass can reach about six times total masses of the tower, rotor, and nacelle, particularly in areas of intense seismic activities (Ishihara ed., 2010). Therefore, seismic responses of wind turbine support structures are severely affected by soil-structure interaction (SSI) (Wolf, 1989; Zhao et al., 2019). An efficient method to account for the effect of SSI is to represent the soil-structure system as a seismic SSI model with a set of springs and dashpots at the soil-foundation interface, substituting the

soil-foundation system. This method has been widely employed for the seismic analysis of wind turbine support structures (Bazeos et al., 2002; Butt and Ishihara, 2012; Stamatopoulos, 2013; Kitahara and Ishihara, 2020). However, by introducing the dashpots, the seismic SSI model is a non-classically damped system and thus classical modal damping models, such as Rayleigh damping model, cannot accurately represent its modal damping properties. Although the modal damping ratios of non-classically damped systems can be theoretically obtained by the complex eigenvalue analysis, they cannot be directly

used for RSM, since complex eigenmodes do not generally coincide with real eigenmodes, which is used in conventional RSM. To overcome this obstacle, Kitahara and Ishihara (2020) have proposed a modal decomposition method to identify equivalent modal damping ratios for the real eigenmodes from the modal damping ratios obtained by the complex eigenvalue analysis, and estimated seismic loadings on wind turbine support structures by conventional RSM using the identified modal damping ratios.

On the other hand, Gao et al. (2019) have recently proposed complex mode superposition RSM for the non-classically damped systems to estimate maximum relative displacements based on the complex eigenmodes. This method employs the modal damping ratios obtained by the complex eigenvalue analysis, and thus it does not require the additional procedure to estimate the equivalent modal damping ratios for the real eigenmodes in Kitahara and Ishihara (2020). In this study, seismic loadings, i.e., maximum shear forces and bending moments, will be newly derived by this method to estimate those on wind

turbine support structures. Gao et al. (2019) have demonstrated this method upon 3- and 6-story shear structures. However, in these structures, highly damped modes are not dominant in their seismic responses, and thus its accuracy in case of highly damped modes have not been investigated. Meanwhile, considering wind turbine support structures, a mode corresponding to the sway motion of the footing is dominant in the shear force acting on footings, and this mode shows significantly large damping ratio due to the soil radiational damping in case of soft soil profiles.

The aim of this study is consequently to further investigate the applicability of complex mode superposition RSM to wind turbine support structures, where highly damped modes are dominant in their seismic responses, and to propose an accurate and efficient method to estimate seismic loadings acting on towers and footings. To achieve this objective, a seismic SSI model of wind turbine support structures is constructed, in which the effect of SSI is considered by a pair of springs and



dashpots in the sway and rocking directions, respectively. Seismic loadings on this model are first derived by Complex mode superposition RSM. To improve the prediction accuracy of the shear force acting on footings, complex mode superposition RSM is then augmented by introducing the upper limit of modal damping ratios. In addition, the bending moment at the hub height due to the mass moment of inertia of rotor and nacelle assembly (RNA) is accounted for as an additional loading by the angular acceleration at the hub height. The proposed method is validated by time history analysis (THA) accounting for different types of foundations and different tower geometries.

## 2 Wind turbine support structures under earthquake

Wind turbine support structures subjected to a horizontal earthquake motion are investigated in this study. As the foundation type, the gravity and piled foundations are considered. In section 2.1, a sway-rocking (SR) model is constructed as the seismic SSI model of wind turbine support structures. In this model, the effect of SSI is considered by a pair of springs and dashpots in the sway and rocking directions, respectively. In addition, an input acceleration response spectrum is defined accounting for the effects of soil amplification and damping correction in Section 2.2. In Section 2.3, seismic loadings on this model is derived by complex mode superposition RSM, and this method is then augmented to improve the prediction accuracy of the shear force acting on footings. The relationship between the structural damping ratios of steel towers and their natural frequencies, as well as additional loadings due to RNA and P–Δ effect are finally summarized in Section 2.4.

### 2.1 Seismic SSI model for wind turbine support structures

The seismic SSI model of wind turbine support structures is constructed as the SR model shown in Fig. 1. Here, x and z express the horizontal and vertical coordinates, respectively. The $k(= 1, \cdots, n-1)$th node represents each degree of freedom (DOF) of the steel tower and footing. In this model, RNA is simplified as a lumped mass at the hub height ($k = n$) and is connected to the tower by using a rigid beam. This simplification has been verified not to influence the prediction accuracy of seismic loadings acting on the tower and footing, excluding that the bending moment at the hub height is underestimated because of no consideration of the mass moment of inertia of RNA (Kitahara and Ishihara, 2020). This bending moment though enables to be obtained as an additional loading by the angular acceleration at the hub height, which is further investigated in Section 2.4. In addition, the tower and footing are modeled by lumped masses and Euler-Bernoulli beam elements. The number of beam elements is set as $n = 27$, which fulfills the requirement in the guidelines for design of wind turbine support structure and foundations by Japan Society of Civil Engineers (JSCE) (Ishihara ed, 2010).

As the foundation type, the gravity and piled foundations are considered. The gravity foundation is typically employed in case of stiff soil profiles, whilst the piled foundation is necessary to be installed in case of soft soil profiles. Regardless of the foundation type, the soil-foundation system is substituted with a pair of springs and dashpots in the sway and rocking directions, respectively, and is connected to the footing bottom. It should be noted that, for simplification, the frequency dependence of the springs and dashpots, cross-coupling between the sway and rocking springs, and mass moment of inertia of the foundation are neglected in this model. The slightly embedded footing with a small embedment ratio, which is defined





as the ratio of the footing depth to width, is employed in this study, and it ensures the representation of SSI by the uncoupled sway and rocking springs and dashpots is a reasonable simplification.

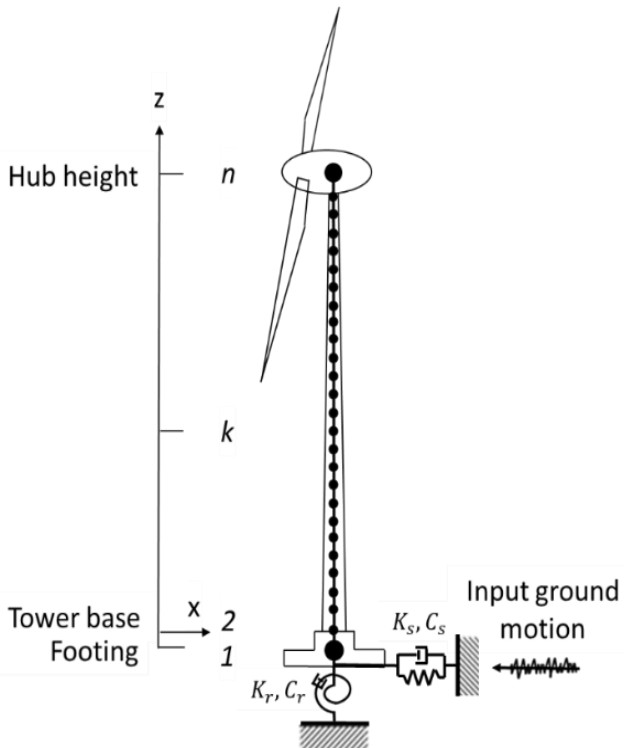

**Figure 1: The seismic SSI model for wind turbine support structures.**

100     The dynamic finite element equation of the seismic SSI model with respect to relative motions can be written as:

$$
\begin{bmatrix} \mathbf{M}_T & \mathbf{0} & \mathbf{0} \\ \mathbf{0} & M_F & 0 \\ \mathbf{0} & 0 & J_F \end{bmatrix} \begin{Bmatrix} \ddot{\tilde{\mathbf{u}}}_T \\ \ddot{u}_F^R \\ \ddot{\theta}_F \end{Bmatrix} + \begin{bmatrix} \mathbf{C}_{TT} & \mathbf{C}_{TF} & -\mathbf{C}_{FF}\mathbf{h} \\ \mathbf{C}_{FT} & C_{FF}+C_s & \mathbf{C}_{FT}\mathbf{h} \\ -\mathbf{h}'\mathbf{C}_{TT} & \mathbf{h}'\mathbf{C}_{TF} & \mathbf{h}'\mathbf{C}_{TT}\mathbf{h}+C_r \end{bmatrix} \begin{Bmatrix} \dot{\tilde{\mathbf{u}}}_T \\ \dot{u}_F^R \\ \dot{\theta}_F \end{Bmatrix}
$$

$$
+ \begin{bmatrix} \mathbf{K}_{TT} & \mathbf{K}_{TF} & -\mathbf{K}_{FF}\mathbf{h} \\ \mathbf{K}_{FT} & K_{FF}+K_s & \mathbf{K}_{FT}\mathbf{h} \\ -\mathbf{h}'\mathbf{K}_{TT} & \mathbf{h}'\mathbf{K}_{TF} & \mathbf{h}'\mathbf{K}_{TT}\mathbf{h}+K_r \end{bmatrix} \begin{Bmatrix} \tilde{\mathbf{u}}_T \\ u_F^R \\ \theta_F \end{Bmatrix} = - \begin{bmatrix} \mathbf{M}_T & \mathbf{0} & \mathbf{0} \\ \mathbf{0} & M_F & 0 \\ \mathbf{0} & 0 & 0 \end{bmatrix} \mathbf{I}\ddot{u}_{g0} , \qquad (1)
$$

where $\mathbf{M}_T$, $\mathbf{C}_{TT}$, and $\mathbf{K}_{TT}$ are the mass, damping, and stiffness matrices of the tower, respectively; $\tilde{\mathbf{u}}_T$ is a column vector of the relative displacement of the tower; $M_F$, $J_F$, $C_{FF}$, and $K_{FF}$ are the mass, moment of inertia, damping coefficient, and stiffness constant of the footing, respectively; $u_F^R$ and $\theta_F$ are column vectors of the relative displacement and rotational angle of the footing; $\mathbf{C}_{TF}$ and $\mathbf{C}_{FT}$ are the damping matrices of the coupling of the tower and footing; $\mathbf{K}_{TF}$ and $\mathbf{K}_{FT}$ are the stiffness

105     matrices of the coupling of the tower and footing; $C_s$ and $K_s$ are the damping coefficient and stiffness constant of the dashpot and spring in the sway direction, respectively; $C_r$ and $K_r$ are those in the rocking direction; $\mathbf{h}$ is a column vector of the height



at each DOF of the tower relative to the footing; **I** is the unit column vector; $\ddot{u}_{g0}$ is the input acceleration time histories at the footing bottom.

For the gravity foundation, the stiffness constants and damping coefficients of the soil-foundation system in Eq. (1), $K_s$, $K_r$, $C_s$, and $C_r$, can be obtained using the cone model as detailed in Architectural Institute of Japan (AIJ) (2006). For the piled foundation, on the other hand, the stiffness constants of the springs in the sway and rocking directions can be calculated by Francis and Randolph models, respectively (Francis, 1964; Randolph, 1981), whereas the damping coefficients of the dashpots can be obtained by Gazetas model (Gazetas and Dobry, 1984). The detailed derivation of these stiffness constants and damping coefficients is given by Ishihara and Wang (2019). Except for the damping coefficients of these dashpots, the damping matrices in Eq. (1) can be obtained by the Rayleigh damping model.

**2.2 Input acceleration response spectrum**

The design acceleration response spectrum is generally defined at the bedrock condition, from which artificial ground motions are generated by considering a given phase property. The input acceleration time histories at the footing bottom $\ddot{u}_{g0}$ used in THA can be then obtained from these artificial ground motions by SHAKE analysis (Schnabel et al., 1972) using the one-dimensional site transfer function. Meanwhile, the input acceleration response spectrum defined at the footing bottom is required in RSM. Several formulae of the acceleration response spectrum are presented, such as Eurocode (2006), the Building Standard Low of Japan (2004), and ASCE (2006), and the following equation defined in the Building Standard Low of Japan (2004) is employed in this study:

$$S_a(T,\zeta) = \begin{cases} a_0 G_s \left\{ 1 + \left( F_\zeta \beta_0 - 1 \right) \dfrac{T}{T_B} \right\} & (0 \leq T < T_B) \\ a_0 G_s F_\zeta \beta_0 & (T_B \leq T < T_C) \\ a_0 G_s F_\zeta \beta_0 \left( \dfrac{T_C}{T} \right) & (T_C \leq T < T_D) \\ a_0 G_s F_\zeta \beta_0 \left( \dfrac{T_C}{T_D} \right)^{K_1} \left( \dfrac{T_D}{T} \right)^{K_2} & (T_D \leq T) \end{cases}, \tag{2}$$

where $T$ and $\zeta$ are the characteristic period and damping ratio; $a_0$ is the peak ground acceleration at the bedrock condition; $G_s$ is the soil amplification factor; $F_\zeta$ is the damping correction factor; $\beta_0$ is the acceleration response magnification ratio for the region where the acceleration response becomes constant; $T_B$, $T_C$, $T_D$, $K_1$, and $K_2$ indicate coefficients representing the shape of the response spectrum. Parameters used for defining the input response spectrum in this study are listed in Table 1. The peak ground acceleration $a_0$ is chosen so that its return period is 475 years as recommended in IEC61400-1 (2019).

**Table 1: Parameters of the input acceleration response spectrum.**

| $a_0$ (m/s$^2$) | $\beta_0$ | $K_1$ | $K_2$ | $T_B$ (s) | $T_C$ (s) | $T_D$ (s) |
|---|---|---|---|---|---|---|
| 3.2 | 2.5 | 1 | 1 | 0.16 | 0.64 | 3.0 |





The design response spectrum at the bedrock condition is typically defined with the damping ratio $\zeta = 0.05$, and the soil amplification factor $G_s$ and damping correction factor $F_\zeta$ will both be one. In the input response spectrum, on the other hand, $G_s$ and $F_\zeta$ should be carefully defined according to the investigated soil profile and structure. In this study, the soil amplification factor $G_s$ is obtained using the one-dimensional site transfer function based on the response spectrum-based method by Okano and Sako (2013). In addition, the damping correction factor $F_\zeta$ employed in this study is expressed as:

$$F_\zeta = \begin{cases} \left(\dfrac{5.2}{0.2 + 100\zeta}\right)^{-0.05T + 0.35\gamma + 0.3} & (\zeta < 0.05) \\[2ex] \left(\dfrac{2}{-3 + 100\zeta}\right)^{0.15 \log_{10} \frac{T}{1.5\gamma} + 0.3} & (\zeta > 0.05) \end{cases}, \tag{3}$$

where $\gamma$ is the quantile value. It should be noted that this equation will be one with the damping ratio $\zeta = 0.05$. Even though the input acceleration time histories are generated from the same design response spectrum, their response spectra largely vary due to differences in phase properties especially in the case with the low damping ratio. Therefore, it is essential to quantify the uncertainty in the response spectra for estimating seismic loadings by RSM with the same reliability level as obtained by THA of several input acceleration time histories. The damping correction factor in Equation (3) is capable to
estimate seismic loadings according to the reliability level by changing the quantile value $\gamma$. More detailed information about the damping correction factor and quantile value is given by Kitahara and Ishihara (2020).

### 2.3 Augmented complex superposition RSM

The equation of motion of the seismic SSI model in Eq. (1) can be converted to a first order matrix equation as (Foss, 1958):

$$\begin{bmatrix} \mathbf{0} & \mathbf{M} \\ \mathbf{M} & \mathbf{C} \end{bmatrix} \begin{Bmatrix} \dot{\tilde{\mathbf{u}}} \\ \ddot{\tilde{\mathbf{u}}} \end{Bmatrix} + \begin{bmatrix} -\mathbf{M} & \mathbf{0} \\ \mathbf{0} & \mathbf{K} \end{bmatrix} \begin{Bmatrix} \tilde{\mathbf{u}} \\ \dot{\tilde{\mathbf{u}}} \end{Bmatrix} = -\begin{Bmatrix} \mathbf{0} \\ \mathbf{MI} \end{Bmatrix} \ddot{u}_{g0}, \tag{4}$$

with

$$\mathbf{M} = \begin{bmatrix} \mathbf{M}_T & \mathbf{0} & \mathbf{0} \\ \mathbf{0} & M_F & \mathbf{0} \\ \mathbf{0} & \mathbf{0} & J_F \end{bmatrix}, \mathbf{C} = \begin{bmatrix} \mathbf{C}_{TT} & \mathbf{C}_{TF} & -\mathbf{C}_{FF}\mathbf{h} \\ \mathbf{C}_{FT} & C_{FF} + C_s & \mathbf{C}_{FT}\mathbf{h} \\ -\mathbf{h}'\mathbf{C}_{TT} & \mathbf{h}'\mathbf{C}_{TF} & \mathbf{h}'\mathbf{C}_{TT}\mathbf{h} + C_r \end{bmatrix}, \mathbf{K} = \begin{bmatrix} \mathbf{K}_{TT} & \mathbf{K}_{TF} & -\mathbf{K}_{FF}\mathbf{h} \\ \mathbf{K}_{FT} & K_{FF} + K_s & \mathbf{K}_{FT}\mathbf{h} \\ -\mathbf{h}'\mathbf{K}_{TT} & \mathbf{h}'\mathbf{K}_{TF} & \mathbf{h}'\mathbf{K}_{TT}\mathbf{h} + K_r \end{bmatrix},$$

and $\tilde{\mathbf{u}} = [\tilde{\mathbf{u}}_T \quad u_F^R \quad \theta_F]^T$.

The complex eigenvalue problem of this first order matrix equation is written as:

$$\left(\lambda_j \begin{bmatrix} \mathbf{0} & \mathbf{M} \\ \mathbf{M} & \mathbf{C} \end{bmatrix} + \begin{bmatrix} -\mathbf{M} & \mathbf{0} \\ \mathbf{0} & \mathbf{K} \end{bmatrix}\right) \mathbf{\Phi}_j = \mathbf{0}, \text{ for } j = 1, 2, \cdots, N, \tag{5}$$

where $\lambda_j$ and $\mathbf{\Phi}_j = \{\lambda_j \mathbf{\phi}_j' \ \mathbf{\phi}_j'\}'$ are the complex eigenvalue and complex eigenvector of the $j$th mode, respectively. Here, $\mathbf{\phi}_j$ represents the $j$th complex mode shape. The eigenvalue $\lambda_j$ and eigenvector $\mathbf{\Phi}_j$ are in complex-conjugate pairs with $\hat{\lambda}_j$ and





$\widehat{\mathbf{\Phi}}_j = \{\lambda_j \widehat{\mathbf{\phi}}'_j \ \widehat{\mathbf{\phi}}'_j\}'$, respectively. Based on the $j$th complex eigenvalue $\lambda_j$, the $j$th natural frequency and modal damping ratio

can be obtained as:

$$\omega_j = |\lambda_j|, \text{and } \zeta_j = -\text{Re}(\lambda_j/|\lambda_j|).\tag{6}$$

On the other hand, the first order matrix equation in Eq. (4) can be also decoupled into the following $N$ single DOF

equations as:

$$\ddot{q}_j + 2\zeta_j \omega_j \dot{q}_j + \omega_j^2 q_j = -\ddot{u}_{g0}, \text{for } j = 1, 2, \cdots, N,\tag{7}$$

where $q_j$ is the displacement response of the single DOF system. By the superposition of the solutions of Eq. (7), the relative

displacement in Eq. (4) can be written as:

$$\widetilde{\mathbf{u}} = \sum_{j=1}^{N} (\mathbf{A}_j q_j + \mathbf{B}_j \dot{q}_j),\tag{8}$$

with

$$\mathbf{A}_j = \omega_j \zeta_j \mathbf{B}_j + i\omega_j \sqrt{1 - \zeta_j} (D_j \mathbf{\phi}_j - \widehat{D}_j \widehat{\mathbf{\phi}}_j), \text{and } \mathbf{B}_j = D_j \mathbf{\phi}_j + \widehat{D}_j \widehat{\mathbf{\phi}}_j,$$

where

$$D_j = \frac{-\mathbf{\phi}'_j \mathbf{M} \mathbf{I}}{2\lambda_j \mathbf{\phi}'_j \mathbf{M} \mathbf{\phi}_j + \mathbf{\phi}'_j \mathbf{C} \mathbf{\phi}_j}, \text{and } \widehat{D}_j = \frac{-\widehat{\mathbf{\phi}}'_j \mathbf{M} \mathbf{I}}{2\widehat{\lambda}_j \widehat{\mathbf{\phi}}'_j \mathbf{M} \widehat{\mathbf{\phi}}_j + \widehat{\mathbf{\phi}}'_j \mathbf{C} \widehat{\mathbf{\phi}}_j}.$$

Maximum relative displacements can be obtained based on the complex complete quadratic combination rule as (Gao et

al., 2019):

$$|\widetilde{\mathbf{u}}|_{max} = \sqrt{\sum_{j=1}^{N} \sum_{l=1}^{N} \{\rho_{jl}^{dd} \mathbf{A}_j \mathbf{A}_l S_{dj} S_{dl} + 2\rho_{jl}^{vd} \mathbf{B}_j \mathbf{A}_l S_{vj} S_{dl} + \rho_{jl}^{vv} \mathbf{B}_j \mathbf{B}_l S_{vj} S_{vl}\}},\tag{9}$$

where $N$ is the highest mode considered in the calculation; $S_{dj}$ and $S_{vj}$ are the relative displacement and relative velocity

response spectra of the $j$th mode, respectively; $\rho_{jl}^{dd}$, $\rho_{jl}^{vd}$, and $\rho_{jl}^{vv}$ are the displacement–displacement, velocity–displacement,

and velocity–velocity correlation coefficients between the $j$th and $l$th modes, respectively. The displacement and velocity

response spectra of the $j$th mode can be approximately obtained from the given acceleration response spectrum by using the

so-called pseudo spectrum transformation as:

$$S_{dj} \cong S_a(T_j, \zeta_j)/\omega_j^2, \text{and } S_{vj} \cong S_a(T_j, \zeta_j)/\omega_j,\tag{10}$$



where $S_a(T_j, \zeta_j)$ is the acceleration response spectrum in Eq. (2) with the $j$th natural period and modal damping ratio.

Moreover, the correlation coefficients $\rho_{jl}^{dd}$, $\rho_{jl}^{vd}$, and $\rho_{jl}^{vv}$ are expressed as:

$$\rho_{jl}^{dd} = \frac{8\sqrt{\zeta_j\zeta_l}(r_{jl}\zeta_j + \zeta_l)r_{jl}^{3/2}}{\left(1 - r_{jl}^2\right)^2 + 4\zeta_j\zeta_l r_{jl}(1 + r_{jl}^2) + 4(\zeta_j^2 + \zeta_l^2)r_{jl}^2}, \tag{11}$$

$$\rho_{jl}^{vd} = \frac{8\sqrt{\zeta_j\zeta_l}(1 - r_{jl}^2)r_{jl}^{1/2}}{\left(1 - r_{jl}^2\right)^2 + 4\zeta_j\zeta_l r_{jl}(1 + r_{jl}^2) + 4(\zeta_j^2 + \zeta_l^2)r_{jl}^2}, \tag{12}$$

$$\rho_{jl}^{vv} = \frac{8\sqrt{\zeta_j\zeta_l}(\zeta_j + r_{jl}\zeta_l)r_{jl}^{3/2}}{\left(1 - r_{jl}^2\right)^2 + 4\zeta_j\zeta_l r_{jl}(1 + r_{jl}^2) + 4(\zeta_j^2 + \zeta_l^2)r_{jl}^2}, \tag{13}$$

where $r_{jl} = \omega_l/\omega_j$ is the natural frequency ratio of the $j$th to $l$th modes.

Maximum seismic loadings at the $k$th node of the seismic SSI model can be finally obtained based on Eq. (9) as:

$$|Q_k|_{max} = \sqrt{\sum_{j=1}^{N} Q_{jk}}, \text{ and } |M_k|_{max} = \sqrt{\sum_{j=1}^{N} M_{jk}}, \tag{14}$$

with

$$Q_{jk} = \sum_{l=1}^{N}\left[\rho_{jl}^{dd}\left\{\sum_{k=i}^{n} A_{jk}A_{lk}S_{dj}S_{dj}m_k\left(\frac{2\pi}{T_j}\right)^2\right\} + 2\rho_{jl}^{vd}\left\{\sum_{k=i}^{n} B_{jk}A_{lk}S_{vj}S_{dj}m_k\left(\frac{2\pi}{T_j}\right)^2\right\}\right.$$
$$\left. + \rho_{jl}^{vv}\left\{\sum_{k=i}^{n} B_{jk}B_{lk}S_{vj}S_{vj}m_k\left(\frac{2\pi}{T_j}\right)^2\right\}\right],$$

and

$$M_{jk} = \sum_{l=1}^{N}\left[\rho_{jl}^{dd}\left\{\sum_{k=i}^{n} A_{jk}A_{lk}S_{dj}S_{dj}m_k(z_n - z_k)\left(\frac{2\pi}{T_j}\right)^2\right\} + 2\rho_{jl}^{vd}\left\{\sum_{k=i}^{n} B_{jk}A_{lk}S_{vj}S_{dj}m_k(z_n - z_k)\left(\frac{2\pi}{T_j}\right)^2\right\}\right.$$
$$\left. + \rho_{jl}^{vv}\left\{\sum_{k=i}^{n} B_{jk}B_{lk}S_{vj}S_{vj}m_k(z_n - z_k)\left(\frac{2\pi}{T_j}\right)^2\right\}\right],$$

where $A_{jk}$ and $B_{jk}$ are the $k$th component of $\mathbf{A}_j$ and $\mathbf{B}_j$, respectively; $m_k$ and $z_k$ are the mass and height of the $k$th node, respectively. It should be noted that Gao et al. (2019) have only derived the maximum relative displacements by complex mode superposition RSM, and the seismic loadings in Eq. (14) are originally derived in this study.





Gao et al. (2019) have demonstrated complex mode superposition RSM upon 3- and 6-story shear structures. Although highly damped and over-damped modes arise at fundamental frequency of these structures, these modes are not dominant in

their seismic responses. In such case, complex mode superposition RSM has been demonstrated to be capable to accurately estimate seismic responses. Nevertheless, this method might underestimate seismic responses of highly damped and over-damped modes, since the velocity–displacement correlation cannot be accurately evaluated by the correlation coefficient in Eq. (12) with such large damping ratios. One of the fundamental modes of wind turbine support structures corresponds to the sway motion of the footing, and this mode shows a significantly large damping ratio due to the soil radiational damping in

the case of soft soil profiles. As detailed later in the numerical verification section, this mode is not dominant in seismic loadings acting on towers and the bending moment acting on towers and footings, whilst it is dominant in the shear force acting on footings used for designing the piled foundations. To prevent the underestimation of the shear force, complex mode superposition RSM is augmented by introducing an empirical formula of the modal damping ratios as:

$$\zeta_j = \max\left(-\mathrm{Re}\left(\lambda_j / |\lambda_j|\right), 0.1\right),\tag{15}$$

where 0.1 is the maximum value of the modal damping ratio based on an engineering judgement. This formula will be

validated by case studies considering different types of foundations and different tower geometries in Section 3.2.

## 2.4 Structural damping of steel towers and additional loadings

Except for the damping coefficients of the dashpots, $C_s$ and $C_r$, the damping matric in Eq. (4) is obtained by the Rayleigh damping model based on the first and second natural frequencies and modal damping ratios. In thus study, the first and second modal damping ratios are assumed to coincide with the structural damping ratio of steel towers, since these two

modes correspond to the sway motion of the tower. The structural damping ratio of steel towers depends on their sizes and an empirical formula was proposed by Oh and Ishihara (2018) as:

$$\zeta_s = \max\left(2.0 e^{-1.3 T_s} + 0.15, 0.2\right) \%,\tag{16}$$

where $\zeta_s$ is the structural damping ratio, and 0.2 % is its minimum value for unlined welded steel stacks as shown in ISO 4354 (2009); $T_s$ is the characteristic period of the fixed foundation model of wind turbine support structures. In this study, the structural damping ratio is obtained by Eq. (16) and is employed as the first and second modal damping ratios.

Moreover, by the simplification of RNA in the seismic SSI model, this model cannot accurately evaluate the bending moment at the hub height due to neglect of the mass moment of inertia of RNA. Hence, an additional loading by the angular acceleration at the hub height is proposed and this loading at the $k$th node of the seismic SSI model is expressed as:

$$M_k^{RNA} = C \times I_y \times \ddot{\theta} \times \left(\frac{z_k}{z_n}\right)^6 = C \times I_y \times \frac{\ddot{u}_n^1 - \ddot{u}_{n-1}^1}{z_n - z_{n-1}} \times \left(\frac{z_k}{z_n}\right)^6, \text{with } C = 0.5,\tag{17}$$





where $I_y$ is the mass moment of inertia of RNA; $\ddot{\theta}$ is the angular acceleration at the hub height; $\ddot{u}_n^1$ is the maximum acceleration of the first mode at the $n$th node; $C$ is the correction factor. The maximum acceleration $\ddot{u}_n^1$ can be obtained as:

$$\ddot{u}_n^1 = \sum_{l=1}^{N}\left[\rho_{1l}^{dd}A_{1n}A_{ln}S_{d1}S_{dl}\left(\frac{2\pi}{T_1}\right)^2 + 2\rho_{1l}^{vd}B_{1n}A_{ln}S_{vj}S_{dl}\left(\frac{2\pi}{T_1}\right)^2 + \rho_{1l}^{vv}B_{1n}B_{ln}S_{v1}S_{vl}\left(\frac{2\pi}{T_1}\right)^2\right]. \tag{18}$$

This additional loading will be validated by comparison with the bending moment at the hub height obtained by THA of the full finite element (FE) model of wind turbine support structures, including the detailed configuration of the rotor and nacelle as shown in Section 3.2. In addition, an additional loading by the P–Δ effect is also proposed in this study as:

$$M_k^{PD} = \sum_{j=k+1}^{n} m_k g\left(u_j - u_k\right), \text{for } k = 1, 2, \cdots, n-1\,, \tag{19}$$

where $g$ is the gravitational acceleration; $u_k$ is the maximum displacement at the $k$th node obtained by Eq. (9).

## 3 Numerical verification and discussion

The proposed method is demonstrated upon numerical examples accounting for different types of foundations and different tower geometries. A typical 2-MW wind turbine is first investigated with different types of foundations, i.e., the gravity and piled foundations. Next, the rated power is varied from 1-MW to 3-MW for the piled foundation supported wind turbine. The outlines of these wind turbine support structures are summarized in Section 3.1. The description of two types of soil profiles used in this study and the input acceleration response spectra for these soil profiles are also presented in this section. Seismic

loadings on the towers and footings are then estimated by the augmented complex mode superposition RSM and compared with the results by THA and the original complex mode superposition RSM in Section 3.2.

### 3.1 Problem statements

Table 2 summarizes the outline of the 2–MW wind turbine and its support structures. The structural damping ratio $\zeta_s$ is estimated by Eq. (16) and is used to obtain the Rayleigh damping model. The footing mass is about six times total masses of

the tower, rotor, and nacelle. The embedded ratio of the footing is assumed to be 0.2. For the piled foundation case, totally eight piles are embedded and extended to the engineering bedrock. The water depth is assumed to be under the pile bottom. Fig. 2 shows the shape of the footing and piled foundation. On the other hand, Table 3 shows the outline of the piled foundation supported wind turbines with the rated power of 1, 1.5, 2, 2.5, and 3-MW constructed based on Xu and Ishihara (2014) and their support structures. Note that, the profiles of the piles are assumed to be same as those shown in Fig.2 for all

these wind turbines. As the soil profiles, typical stiff and soft soil profiles, namely Soil type I and II shown in AIJ (2006) are considered in this study. The gravity foundation is used for Soil type I whilst the piled foundation is used for Soil type II.





Table 4 presents the description of one-dimensional layered soil models for these two soil profiles. Table 5 summarizes the stiffness constants and damping coefficients of the soil-foundation systems for these two soil profiles.

**Table 2: The outline of the 2-MW wind turbine and its support structures.**

| | |
|---|---|
| Rotor diameter (m) | 83 |
| Tower height (m) | 67 |
| Rotor and nacelle mass (kg) | 112000 |
| Tower mass (kg) | 165100 |
| Tower top diameter (m) | 2.34 |
| Tower top thickness (mm) | 13 |
| Tower bottom diameter $\phi$ (m) | 4.23 |
| Tower bottom thickness (mm) | 35 |
| Structural damping ratio $\zeta_S$ (%) | 0.2 |
| Footing width $B = B_1 = B_2$ (m) | 16 |
| Footing depth $H$ (m) | 3 |
| Footing mass (kg) | 1551170 |
| Pile diameter $\phi_P$ (m) | 1.5 |
| Pile distance $S$ (m) | 6.5 |
| Pile length $L$ (m) | 22 |
| Number of piles in the $x$–direction | 3 |
| Number of piles in the $y$–direction | 3 |
| Total number of piles | 8 |
| Young's modulus of the pile (kN/m$^2$) | 22800000 |
| Density of the pile (kg/m$^3$) | 2446.5 |

**Table 3: The outline of the piled foundation supported wind turbines with different rated powers.**

| Item | Description | | | | |
|---|---|---|---|---|---|
| Rated power (MW) | 1 | 1.5 | 2 | 2.5 | 3 |
| Rotor diameter (m) | 56 | 72 | 83 | 92 | 95 |
| Tower height (m) | 56 | 60 | 67 | 72 | 75 |
| Rotor and nacelle mass (kg) | 60200 | 89400 | 112000 | 130500 | 136000 |
| Tower mass (kg) | 90300 | 107500 | 165100 | 179000 | 187200 |
| Turbine total mass (kg) | 1505000 | 196900 | 277100 | 309500 | 323900 |
| Footing width (m) | 13 | 14 | 16 | 16 | 16 |
| Footing depth (m) | 2 | 2.5 | 3 | 3 | 3 |
| Footing mass (kg) | 813422.3 | 989678.8 | 1551170 | 1551170 | 1551170 |
| Mass ratio | 0.185 | 0.199 | 0.179 | 0.200 | 0.209 |
| Natural frequency of the first mode (Hz) | 0.483 | 0.475 | 0.404 | 0.378 | 0.371 |
| Structural damping ratio (%) | 0.285 | 0.280 | 0.230 | 0.214 | 0.210 |



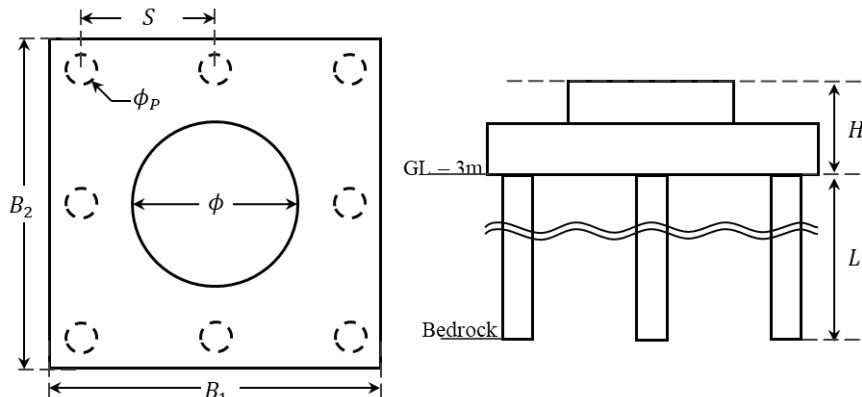

**Figure 2: The shape of the footing and piled foundation.**

**Table 4: The description of one-dimensional layered soil models.**

**(a) Soil type I**

| Layer No. | Depth $D$ (m) | Density $\rho$ (t/m$^3$) | S-wave Velocity $V_S$ (m/s) | P-wave Velocity $V_p$ (m/s) | Soil type |
|---|---|---|---|---|---|
| 1 | 3.0 | 1.7 | 130 | 320 | Sand |
| 2 | 5.7 | 1.8 | 340 | 720 | Sand |
| 3 | 10.0 | 1.7 | 280 | 720 | Clay |
| 4 | 17.4 | 1.9 | 380 | 1980 | Sand |
| Bedrock | — | 2.1 | 510 | 1980 | Rock |

**(b) Soil type II**

| Layer No. | Depth $D$ (m) | Density $\rho$ (t/m$^3$) | S-wave Velocity $V_S$ (m/s) | P-wave Velocity $V_p$ (m/s) | Soil type |
|---|---|---|---|---|---|
| 1 | 4.5 | 1.8 | 90 | 1360 | Clay |
| 2 | 10.0 | 1.6 | 150 | 1560 | Sand |
| 3 | 17.0 | 1.8 | 210 | 1560 | Sand |
| 4 | 18.5 | 1.7 | 150 | 1560 | Clay |
| 5 | 25.0 | 1.8 | 260 | 1560 | Sand |
| Bedrock | — | 1.8 | 400 | 1700 | Rock |

**Table 5: Stiffness constants and damping coefficients for soil-foundation systems.**

| | Sway | | Rocking | |
|---|---|---|---|---|
| | Stiffness constant (N/m) | Damping coefficient (Nsec/m) | Stiffness constant (Nm/rad) | Damping coefficient (Nmsec/rad) |
| Soil type I | $8.56 \times 10^8$ | $2.07 \times 10^7$ | $5.74 \times 10^{11}$ | $7.04 \times 10^8$ |
| Soil type II | $7.90 \times 10^8$ | $3.02 \times 10^7$ | $4.03 \times 10^{11}$ | $1.02 \times 10^9$ |

Fig. 3 shows the input acceleration response spectra at the footing bottom obtained for both soil profiles, together with the design acceleration response spectrum at the bedrock condition. In these response spectra, the damping ratio is assumed to be $\zeta = 0.05$, the soil amplification factor $G_s$ is obtained by Okano and Sako (2014), and the damping correction factor is assumed to be $F_\zeta = 1$. Compared with the design response spectrum at the bedrock condition, the input response spectrum at





the footing bottom is amplified in the short period range less than 0.5 s for Soil type I, whilst it is amplified in the long
period range larger than 0.5 s for Soil type II. Note that, the response spectra shown in Fig. 3 are not directly employed in
RSM, but instead, $F_\zeta$ is obtained for each modal damping ratio by Eq. (3) and is multiplied with these response spectra in the
RSM procedure.

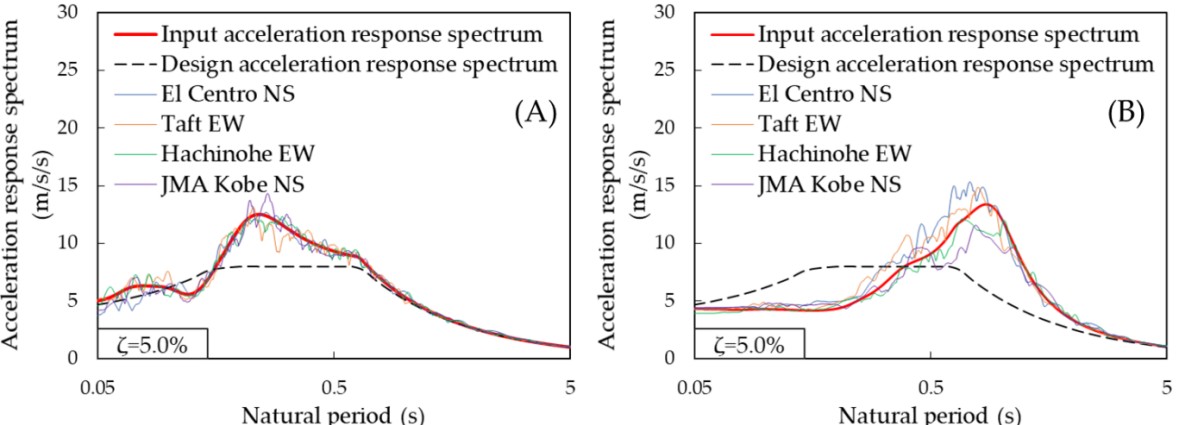

**Figure 3: Input acceleration response spectra at the footing bottom: (A) Soil type I, (B) Soil type II.**

On the other hand, 15 artificial ground motions are generated considering different phase properties from the design
response spectrum. The four ground motions, namely El Centro NS, Taft EW, Hachinohe EW, and JMA Kobe NS show
phase properties of these famous observed earthquake records (Building Performance Standardization Association; Japan
Meteorological Agency), and the other 11 ground motions show random phase properties. Input acceleration time histories at
the footing bottom $\ddot{u}_{g0}$ are then obtained from these artificial ground motions for both soil types by DYNEQ (Yoshida and
Suetomi, 1995), which allows similar analysis as SHAKE (Schnabel et al., 1972), based on the equivalent linearization
method. Note that, the shear strain is less than 1 % for both Soil type I and II, and therefore the equivalent linearization
method is applicable for both cases. The acceleration response spectra of the input acceleration time histories are also given
in Fig. 3 for the case with the phase properties of the four observed earthquake records. It can be seen the response spectra of
these input acceleration time histories are good agreement with the input acceleration response spectrum, while they vary
due to differences in the phase properties especially for Soil type II. Therefore, seismic loadings on wind turbine support
structures obtained by THA using these input acceleration time histories will also vary. No matter what kind of method is
employed in the seismic analysis, such uncertainty in seismic loading estimation should be carefully addressed, and in the
proposed method, it is achieved by the quantile value $\gamma$ in the damping correction factor.

**3.2 Seismic loading estimation by augmented complex mode superposition RSM**

Seismic loadings acting on the towers and footings are estimated by the proposed method. In this study, the quantile value in
the damping correction factor is set to be $\gamma = 0.5$ to estimate mean profiles of maximum seismic loadings. The results are



compared with mean values of the results obtained by THA using the 15 input acceleration time histories. It should be noted that, while only the mean profiles are considered in this study, the proposed method is also capable to estimate the seismic loadings according to the desired reliability level by changing the quantile value, and more detailed information is given in Kitahara and Ishihara (2020).

       Table 6 shows the first five natural frequencies and modal damping ratios of the 2-MW wind turbine support structures

obtained by the complex eigenvalue analysis. In addition, the corresponding modal participation functions $D_j\boldsymbol{\phi}_j$, for $j = 1,\cdots,5$, are obtained and illustrated in Fig. 4. Considering up to the fifth mode fulfils the criteria of Model Code for Concrete Chimneys (CICIND, 2011). It can be seen that a very large damping ratio larger than 40 % arises at the third mode in case with Soil type II, while all the modal damping ratios are less than 10 % in case with Soil type I. As can be seen in the modal participation function, the third mode in case with Soil type II corresponds to the sway motion of the footing, and the large

damping ratio arisen at this mode is caused by soil radiational damping. In the proposed method, this modal damping ratio is substituted with 10 % by Eq. (15) as provided in the parentheses in Table 6 to avoid the underestimation of the shear force acting on the footing.

**Table 6: Modal properties of 2-MW wind turbines.**

|  | Soil type I | | Soil type II | |
|---|---|---|---|---|
|  | Natural frequency (Hz) | Damping ratio (%) | Natural frequency (Hz) | Damping ratio (%) |
| 1st | 0.404 | 0.2 | 0.404 | 0.2 |
| 2nd | 3.021 | 0.2 | 3.003 | 1.5 |
| 3rd | 8.850 | 0.8 | 3.534 | 40.8 (10.0) |
| 4th | 11.765 | 8.5 | 8.929 | 0.8 |
| 5th | 17.241 | 1.1 | 17.241 | 1.1 |

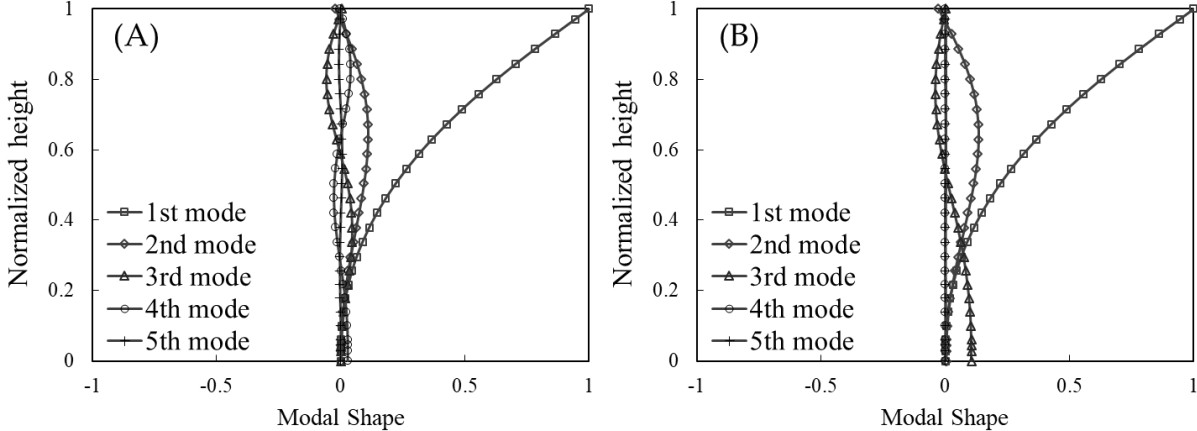

**Figure 4: Real parts of complex modal participation functions: (A) Soil type I, (B) Soil type II.**



Fig. 5 illustrates mean profiles of the maximum shear forces and bending moments acting on the towers estimated by the proposed method, together with those by the original complex mode superposition RSM by Gao et al. (2019), namely CRSM, and mean values of the results obtained by THA using the 15 input acceleration time histories. It can be seen that CRSM and the proposed method provide similar profiles and estimate the seismic loadings which are favourable agreement with the mean values of the results by THA, regardless of the foundation type and soil profile. As investigated in Kitahara and Ishihara (2020), the first and second modes, corresponding to the sway motion of the tower, are dominant in the seismic loadings on the towers. The damping ratios of these modes are small; hence, the proposed method degrades into CRSM, and complex mode superposition RSM works well to estimate seismic loadings which are close to the THA results.

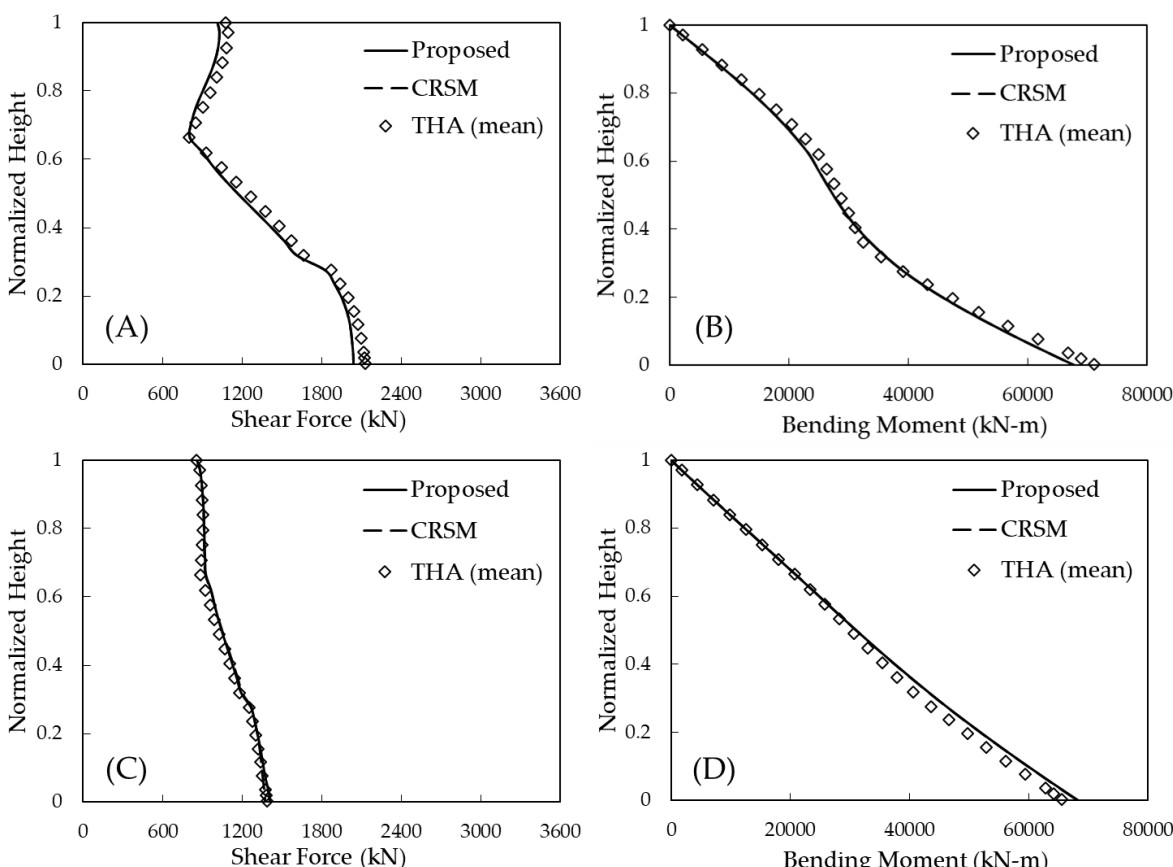

**Figure 5: Vertical profiles of seismic loadings on towers: (A) Shear force on Soil type I, (B) Bending moment on Soil type I, (C) Shear force on Soil type II, (D) Bending moment on Soil type II.**

On the other hand, Fig. 6 shows a comparison of mean values of the seismic loadings acting on the footings obtained by the proposed method as well as CRSM and THA. It can be seen that, except for the shear forces in case with Soil type II, all the methods provide similar results. As shown in Kitahara and Ishihara (2020), the first and second modes are dominant in the bending moment on the footings as same as in the seismic loadings on the towers. Meanwhile, the fourth mode in case





with Soil type I and third mode in case with Soil type II, corresponding to the sway motion of the footing, are dominant in the shear force on the footings. In case with Soil type I, the fourth modal damping ratio is less than 10 %, and in such case, it is found complex mode superposition RSM is capable to estimate seismic loadings similar to the THA results. In case with Soil type II, conversely, the third modal damping ratio is larger than 10 %, and CRSM significantly underestimates the shear force. In the proposed method, this modal damping ratio is hence substituted with 10 % to avoid such underestimation, and the estimated shear force is good agreement with the THA result, demonstrating that the proposed upper limit of modal damping ratios of 10 % is a reasonable choice.

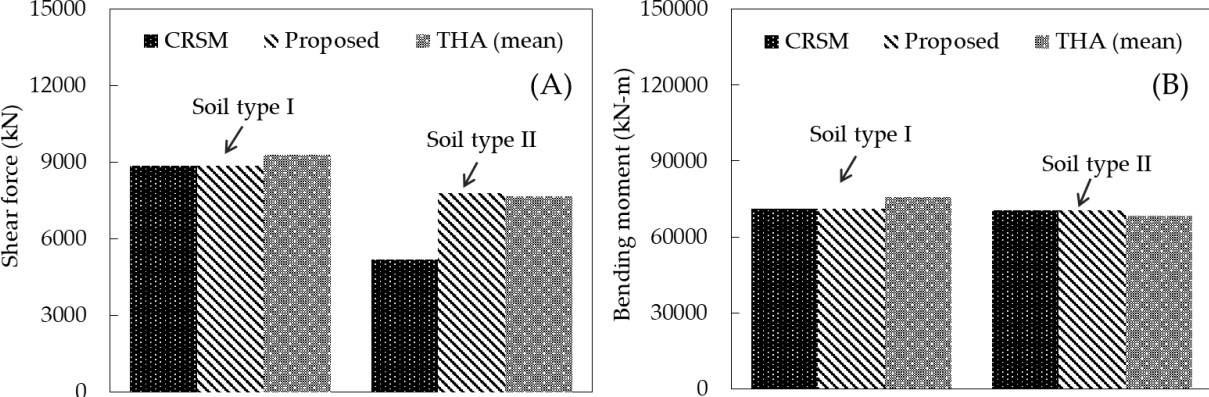

**Figure 6: Seismic loadings on the footings: (A) Shear forces, (B) Bending moments.**

Table 7 summarizes prediction errors in the seismic loadings at the tower base and footing by the proposed method and CRSM compared with the results by THA. As can be seen, the prediction accuracy of the proposed method is quite well and prediction errors are less than 6 % for all cases regardless of the foundation type and soil profile, whilst CRSM significantly underestimates the shear force on the footing and the prediction error is larger than 30 %.

**Table 7: Prediction errors (%) in the seismic loadings.**

|  |  | Shear force | | Bending moment | |
|---|---|---|---|---|---|
|  |  | Tower base | Footing | Tower base | Footing |
| Soil type I | CRSM | -4.21 | -5.08 | -4.90 | -5.99 |
|  | Proposed | -4.21 | -5.08 | -4.90 | -5.99 |
| Soil type II | CRSM | -1.13 | -32.30 | 3.73 | 3.25 |
|  | Proposed | 1.77 | 1.39 | 3.76 | 3.32 |

In addition, additional loadings by the angular acceleration at the hub height are obtained by Eq. (17) so as to avoid the underestimation of the bending moments at the hub height caused by the simplification of RNA. The mass moment of inertia of RNA is assumed to be $I_y = 3814.3$ kN. Fig. 7 illustrates a comparison of mean vertical profiles of the maximum bending moments on the towers by the proposed method, together with the results by THA of the full FE model of the wind turbine





support structures, including the detail configuration of the rotor and nacelle. It can be seen that those two profiles match well and the proposed additional loadings are capable to accurately evaluate the bending moments at the hub height due to the mass moment of inertia of RNA. At the same time, the additional loadings by the P–Δ effect are also evaluated by Eq. (18). Table 8 summarizes the estimated additional loadings at the tower base $M_2^{PD}$ and the ratios of these additional loadings to the bending moments at the tower base $M_2^{PD}/|M_2|_{max}$. It can be seen $M_2^{PD}/|M_2|_{max}$ is less than 3 % for both soil profiles,

thus the P–Δ effect can be ignored in the practical use.

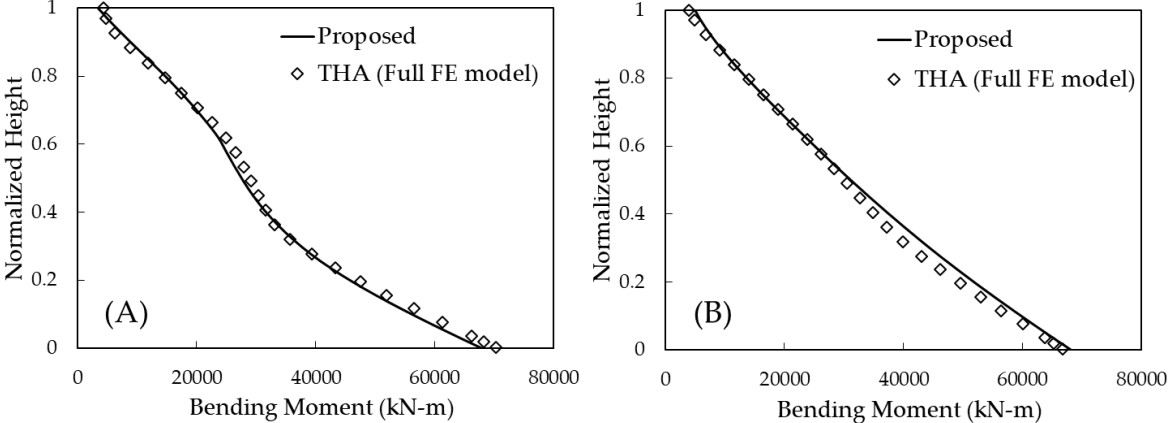

**Figure 7: Vertical profiles of bending moments on the towers considering the additional loadings: (A) Soil type I, (B) Soil type II.**

**Table 8: Additional loadings by the P–Δ effect at the tower base.**

|  | $M_2^{PD}$ (kN–m) | $M_2^{PD}/|M_2|_{max}$ |
|---|---|---|
| Soil type I | 1313 | 1.8 % |
| Soil type II | 1538 | 2.2 % |

        The accuracy of the proposed method is then further verified considering different tower geometries. Fig. 8 illustrates a

comparison of the shear forces and bending moments at the 1/2 height, tower base, and footing estimated by the proposed method and the those mean values obtained by THA using the 15 input acceleration time histories. The results obtained by CRSM are also plotted in this figure. It can be seen that the predicted seismic loadings on the towers by both the proposed method and CRSM show favourable agreement with those by THA. It is because the first and second modal damping ratios are less than 10 % for all cases and the proposed method degrades into CRSM. The same conclusion can be obtained for the

bending moments on the footings. The shear forces on the footings, on the other hand, are significantly underestimated by CRSM because very large damping ratios arise at the mode corresponding to the sway motion of the footing, while the proposed method accurately estimates the shear forces, demonstrating the proposed upper limit of modal damping ratios of 10 % is applicable regardless of the tower geometry.





**330  Figure 8: The comparison of seismic loadings by RSM and THA for different tower geometries: (A) Shear forces on towers, (B) Bending moments on towers, (C) Shear forces on footings, (D) Bending moments on footings.**

**4 Conclusion**

In this study, the seismic SSI of wind turbine support structures is investigated using RSM. The non-classically damped seismic SSI model of wind turbine support structures is constructed, and its seismic loadings are analytically derived from

complex mode superposition RSM. Complex mode superposition RSM is herein augmented by introducing the upper limit of modal damping ratios of 10 % to improve the prediction accuracy of the shear force acting on the footing. Moreover, the mass moment of inertia of RNA and P–Δ effect are also considered as additional loadings analytically derived from complex mode superposition RSM. The proposed method is demonstrated upon numerical examples accounting for different types of foundations and different tower geometries. These examples demonstrate that the proposed method is capable to accurately



and efficiently estimate seismic loadings on wind turbine support structures. Some conclusions of this study are summarized below;

1. Seismic loadings, i.e., the maximum shear force and bending moment, are analytically derived from complex mode superposition RSM. This method is based on the complex eigenmodes obtained by the complex eigenvalue analysis, thus it is applicable to the non-classically damped systems.

2. A highly damped mode arises at fundamental frequency of wind turbine support structures in case with soft soil profiles, and CRSM significantly underestimates the shear force acting on the footing, which is important for designing piled foundations. Thus, CRSM is augmented by introducing the upper limit of modal damping ratios of 10 % to avoid such underestimation.

3. The proposed method is validated by comparison with time history analysis (THA) accounting for different types of
foundations and different tower geometries, and seismic loadings on the towers and footings estimated by the proposed method show favourable agreement with the THA results.

**Acknowledgements**

This research was carried out as a part of the project funded by ClassNK, J-POWER, Shimizu Corporation, Toshiba Energy Systems & Solutions Corporation, and MHI Vestas Offshore Wind Japan. The authors express their deepest gratitude to the
concerned parties for their assistance during this study.

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
