# Peer review of "Seismic soil-structure interaction analysis of wind turbine support structures using augmented complex mode superposition response spectrum method"

_Wind Energy Science, 2021_

## Author Response (AR1)

Answer to the comments of reviewer and corresponding modifications

Dear Reviewer #1,

    The authors thank to the valuable comments from the reviewer and we modified our paper as your suggestion.

| No. | Comment | Answer |
|---|---|---|
| 1 | By reading the manuscript, the reviewer is wondering which exactly is the scientifically novel part of the study presented by the authors. If the reviewer is not mistaken, the authors applied an existing technique for RSM for the seismic analysis of wind turbine support structures. This method has been successfully applied for building type structures but not for wind turbines. So, one difference between the current study and the already published one can be seen in the structures (building vs wind turbine) that this method have been applied. The other difference is that the authors applied a threshold on some excessive values that can be derived for modal damping ratio. This correction is based on an empirical formula (Eq. 15) that does not necessarily come from the authors. Hence, in other words, one can say that the current manuscript reflects the application of an existing method in order to calculate the response of a wind turbine subjected to seismic forces while the use of existing formula is also adopted herein to fix some excessive damping ratio values. The reviewer has nothing to say against that this application and the results seem to be | As the reviewer mentioned, the present work is rooted on the existing technique for RSM, however a novel method, the augmented complex mode superposition RSM, is developed originally by the authors to extend the applicability of the technique for seismic loading estimate of wind turbine support structures. The novel contributions are mainly three folds:
 (i) The already published study, in which the complex mode superposition RSM was proposed, was aimed at estimating peak values of story drifts of building type shear structures, and only the maximum displacement of the multi-DOF system was analytically derived. On the other hand, the seismic design of wind turbine support structures requires the maximum shear force and bending moment acting on the tower and footing. Thus, they are analytically derived in this study based on the framework of the complex mode superposition.
 (ii) While the modal damping rations are calculated by solving a complex eigenvalue problem in the complex mode superposition RSM, an empirical formula (Eq. 16 in the revised manuscript) is proposed in this study to substitute a given allowable damping ratio for the excessive modal damping ratios. This formula is not an existing one but is rather proposed by the authors based on the parametric study in this study, in which different modal damping ratios from 6 % to 58 % are considered for the most dominant mode on the shear force on the footing, by changing the tower geometries and soil conditions. It is found that 0.1 is a reasonable choice for the allowable damping ratio to improve the prediction accuracy of the shear force on the footing.
 (iii) To consider the contribution of the mass moment of inertial of the rotor and nacelle assembly and the p- |

| | | |
|---|---|---|
| | quite promising since a high similarity was found between the results from the currently applied method and the THA. However, the reviewer is bit reserved about the overall novelty of the current study. According to the reviewer's opinion, the current manuscript fits better to an application paper (or technical note) rather than an original research article. The authors are kindly asked to provide their point of view for this issue. However, it is also an issue that the Editor can have a word. | $\Delta$ effect to the bending moment on the tower, Eqs. (17-19) are derived by the authors based on the framework of the complex mode superposition. These three contributions are necessary to analytically estimate the seismic loadings on wind turbine support structures. In the revised manuscript, aforementioned original contributions are clearly stated to emphasize the novelty of the present work in the lines 64-74 of the introduction part. |
| 2. | Can the authors describe the origin of the empirical formula that they used to define this threshold for the modal damping ratio? Especially, the reviewer is interested in the 0.1 value that is included in the formula. Is this value based on engineering judgement? | This empirical formula is proposed in this study based a parametric study varying the damping ratio of the most dominant mode on the shear force on the footing from 6 % to 58 %. In the revised manuscript, the 0.1 value in Eq. (16) is replaced by $\zeta_{thr}$, i.e., the threshold value for the allowable damping ratio, and it is then emphasized in the lines 193-196 that 0.1 is found as a reasonable choice of $\zeta_{thr}$ by the parametric study. |
| 3. | The damping ratio that was found for the 3rd mode and Soil Type was equal to 40.8. Indeed, it is an excessive number. However, this manuscript describes a specific case, for which this high value was calculated. There is a chance that the application of the current method for another case (different soil type, different wind turbine supporting structure etc.) will lead to another value for the damping ratio, for example, 15%. So, what should someone do in this case? Which is the limit of the damping ratio over which the substitution | Based on the reviewer's comment, the authors added additional cases with different soil conditions to the parametric study in the revised manuscript to vary the damping ratio of the most dominant mode on the shear force acting on the footing from 6 % to 58 %. The figure below (Fig. 8 in the revised manuscript) shows that the threshold value $\zeta_{thr} = 0.1$ results in accurate estimates of the shear force on the footing for all case while no consideration of the threshold or $\zeta_{thr} = 0.15$ result in underestimates and $\zeta_{thr} = 0.05$ results in overestimates. It is thus concluded in this study that 0.1 is a reasonable choice for the limit of the damping ratio over which the substitution should be take place. This parametric study is summarized in the lines 313-337 in the revised manuscript. |

should take place? In other words, the method that is presented by the authors should have somehow a more general validity and should not be highly case-specific and highly dependent on engineering judgment.

[Figure]

Figure 8: Normalized shear force on the footing for different modal damping ratios.

| 4. | The reviewer is a bit confused about the earthquake records that were artificially generated and used for the THA. Especially, Fig. 3 shows, among others, the response spectra of four recorded (natural) strong ground motions. So, did the authors used recorded ground motions for the THA or did they used artificial ones? Or both of them? And, why did the authors choose to show the response spectra of the existing ground motions and not of the artificially generated ones? | In the entire manuscript, the authors used artificially generated ground motions obtained from the design spectra. In Fig. 3, not the response spectra of recorded strong ground motions but the artificially generated ground motions having the phase properties of these recorded strong ground motions are illustrated. In the revised manuscript, the captions in Fig. 3 is modified e.g., "El Centro NS phase" to avoid confusion. |
| 5. | By the beginning of chapter 3, the authors describe one wind turbine (2 MW) and two foundation solutions (gravity and piles). Then, rated power was varying – hence, different foundations (i.e., different footings) as well as different characteristics of the overall wind turbine structure were considered. However, it is not clear at all for which of the | For clarifying which case each figure corresponds to, Section 4 in the revised manuscript (that corresponds to Section 3 in the original one) is divided into two subsections, where the former summarizes the results on the 2-MW wind turbine with two different types of foundation, while the latter focuses on the results of the parametric study with different tower geometries and soil conditions. |

| | aforementioned cases the authors present results. For example, Fig. 5 provide results of shear forces and bending moments along the height. However, to which of the aforementioned cases do these results correspond? The same is valid for all the results that the authors present. | |
| --- | --- | --- |

Dear Reviewer #2,

The authors thank to the valuable comments from the reviewer and we modified our paper as your suggestion.

| No. | Comment | Answer |
|---|---|---|
| 1 | The novelty of the newly proposed augmented complex mode superposition response spectrum method is suggested to clarify by comparing with the previous methods in Section 2. | To clarify the difference between the previous method and the proposed method, the authors provided a new section (Section 3 with the title: Augmented complex mode superposition RESM) in the revised manuscript, where Section 3.1 gives a brief review of the previous method, while Section 3.2 describes its extension to the proposed method. |
| 2. | The title of Section 2 "Wind turbine support structures under earthquake" is not proper. This section mainly introduces the methodology and the proposed new method in this study. | In the revised manuscript, the authors divided Section 2 into two distinct sections. The former containing the current subsections 2.1 and 2.2 keeps the title "Wind turbine support structures under earthquake", while the latter introducing the proposed method is named as "Augmented complex mode superposition RSM". |
| 3. | In lines 122 and 123. "Building Standard Low of Japan" seems to a spelling mistake of "Building Standard Law of Japan". | Thank you for raising this. The authors corrected these spelling mistakes in the revised manuscript. |
| 4. | This paper only provides the dynamics equation for SSI system model, but there is no consideration of the P-Delta effect which should be accounted as a dominant factor for seismic analysis. | In this study, the contribution of the p-delta effect on the bending moments acting on wind turbine support structures are considered as an additional loading by the proposed formula, Eq. (19), and is demonstrated on the 2-MW wind turbine example in Section 4.1, the lines 300-308. |
| 5. | Regarding the damping ratio, P–$\Delta$ effect is proposed as in Eq. (19). However, there no detailed information about the applying of this equation on the structural damping. | As mentioned above, Eq. (19) is not applied on the structural damping but on estimating the contribution of the p-delta effect on the bending moment acting on the tower. |
| 6. | | As stated in the line 215, the first mode damping ratio |

| | It is noted in Table 6 that the first mode damping ratio is only 0.2%, which is far smaller than typical steel structures. Please clarify the computation of this damping and the reasonability. | of 0.2 % is computed by Eq. (2), which is derived in Oh and Ishihara (2018) based on several experiments on onshore and offshore wind turbines with different rated powers. |
|---|---|---|